# Tetrachromatic vision-inspired neuromorphic sensors with ultraweak ultraviolet detection

Ting Jiang[1,10], Yiru Wang[2,10], Yingshuang Zheng[1], Le Wang[2], Xiang He[2], Liqiang Li [1,3], Yunfeng Deng[4], Huanli Dong [5], Hongkun Tian [6] ✉, Yanhou Geng[4], Linghai Xie [2], Yong Lei [7], Haifeng Ling [2] ✉, Deyang Ji [1,3] ✉ & Wenping Hu [3,8,9]

Sensing and recognizing invisible ultraviolet (UV) light is vital for exploiting advanced artificial visual perception system. However, due to the uncertainty of the natural environment, the UV signal is very hard to be detected and perceived. Here, inspired by the tetrachromatic visual system, we report a controllable UV-ultrasensitive neuromorphic vision sensor (NeuVS) that uses organic phototransistors (OPTs) as the working unit to integrate sensing, memory and processing functions. Benefiting from asymmetric molecular structure and unique UV absorption of the active layer, the as fabricated UV-ultrasensitive NeuVS can detect 370 nm UV-light with the illumination intensity as low as 31 nW cm$^{-2}$, exhibiting one of the best optical figures of merit in UV-sensitive neuromorphic vision sensors. Furthermore, the NeuVS array exibits good image sensing and memorization capability due to its ultrasensitive optical detection and large density of charge trapping states. In addition, the wavelength-selective response and multi-level optical memory properties are utilized to construct an artificial neural network for extract and identify the invisible UV information. The NeuVS array can perform static and dynamic image recognition from the original color image by filtering red, green and blue noise, and significantly improve the recognition accuracy from 46 to 90%.

With the collaborative cooperation of eyes and brain, visual information is detected and pre-processed by the retina and then transmits to the visual cortex for further complex perceptual processing[1,2]. Neuromorphic vision sensor (NeuVS) can emulate the in-sensor computing operations of visual perception system through integrating functions of photosensor and artificial synapse in one device[3]. This highly compact all-in-one sensor device could simplify the circuit in artificial vision system, improve information processing efficiency and reduce

[1]Tianjin Key Laboratory of Molecular Optoelectronic Sciences, Department of Chemistry, Institute of Molecular Aggregation Science, Tianjin University, Tianjin 300072, China. [2]State Key Laboratory of Organic Electronics and Information Displays & Institute of Advanced Materials (IAM), Nanjing University of Posts & Telecommunications, Nanjing 210023, China. [3]Haihe Laboratory of Sustainable Chemical Transformations, Tianjin 300192, China. [4]School of Materials Science and Engineering, Tianjin University, Tianjin 300072, China. [5]Beijing National Laboratory for Molecular Sciences, Key Laboratory of Organic Solids, Institute of Chemistry, Chinese Academy of Sciences, Beijing 100190, China. [6]State Key Laboratory of Polymer Physics and Chemistry, Changchun Institute of Applied Chemistry, Chinese Academy of Sciences, Changchun 130022, China. [7]Fachgebiet Angewandte Nanophysik, Institut für Physik & IMN MacroNano, Technische Universität Ilmenau, Ilmenau 98693, Germany. [8]Tianjin Key Laboratory of Molecular Optoelectronic Sciences, Department of Chemistry, School of Science, Tianjin University. Collaborative Innovation Center of Chemical Science and Engineering, Tianjin 300072, China. [9]Joint School of National University of Singapore and Tianjin University, Fuzhou 350207, China. [10]These authors contributed equally: Ting Jiang, Yiru Wang. ✉e-mail: hktian@ciac.ac.cn; iamhfling@njupt.edu.cn; jideyang@tju.edu.cn

power consumption[4–8]. In retina, photoreceptor cell (PRC, i.e., cone cells) could detect and convert light into electrical signals to stimulate biological perception processes. Depending on the spectral sensitivity, typically, human retina contains three types of PRC to perceive visible light, forming trichromatic vision[9–13]. As an invisible part of solar radiation, ultraviolet (UV) radiation plays an important role in sterilization, identification, perspective, medical treatment and health[14]. Unlike humans, several species of insects (e.g., butterflies, fish, and birds) can see UV light, endowing them the ability of tetrachromatic vision. For example, butterflies can distinguish stamen and pistil from petal of plants in an UV light pattern for eating targeted nectar accurately[15]. Despite the emerging importance of tetrachromatic vision in advanced artificial visual system, relatively little attention has been paid to sensing and processing of UV light[7,16,17].

On the other hand, UV light with the wavelength of 320–400 nm is supposed to have long wave black spot effect for strong penetration and is harmful to the retina, especially damaging the cornea[2,18]. In view of this, exploring the perception UV range of neuromorphic vision sensors is very necessary for early warning of ultraviolet radiation. In actual ultraviolet detection, the intensity of the target is usually too weak, and the large numbers of gas molecules or dust in the environment can strongly absorb and scattering the ultraviolet light, giving rise to a UV signal much weaker to be detected[19]. Moreover, due to the uncertainty of the natural environment, the existence of other light wavelengths as the noise would interfere with the detection. Organic phototransistors (OPTs) represent a viable approach for developing NeuVS that integrates light information sensing-memory-processing capabilities[20–22]. Due to tailorable molecular properties, this three-terminal optoelectronic device can effectively modulate the channel carriers, electric field distribution, and photocurrent for realizing ultraweak UV light detection[23–27]. To date, however, the cases of OPTs with UV detection sensitivity of a few $\mu W\,cm^{-2}$ level or even lower detection were only reported in single crystal devices, precise manipulation of single crystals is still a daunting challenge for large-scale fabrication and more complex integration applications[28,29]. To this end, thin-film OPTs offer a reliable platform for maximizing the UV sensitivity of organic semiconductors through grain boundaries, interface traps, and the compatibility of organic devices with microelectronic technologies.

Here, 2-hexylthieno[4,5-b][1] benzothieno[3,2-b][1] benzothiophene (BTBTT6-syn) with asymmetric molecular structure and unique UV absorption is selected as the active layer to prepare UV-sensitive NeuVS. The as-fabricated thin-film OPTs show ultraweak UV detection with light intensity as low as $31\,nW\,cm^{-2}$. What's more, the best optical figures of merit, including photosensitivity (P) closing to $10^6$, the photoresponsivity (R) as high as $10^7\,A\,W^{-1}$, and the specific detectivity (D*) above $10^{17}$ Jones, presenting one of the highest values among all organic photodetectors. In addition to UV-ultrasensitive property, the OPTs can also emulate the biological synapses for light signal sensing, memory and pre-processing. The synaptic OPTs show reliable long-term potentiation (LTP) functions with the best retention time up to 20,000 s. An artificial neural network (ANN) with the NeuVS cores is simulated to perform static and dynamic image pre-processing by extracting UV information from the original optical image. With filtering red, green, and blue (RGB) noise, the recognition rate on the handwritten digit is improved from 46 to 86%. By modulating the trap density at the dielectric layer interface, the recognition rate of >90% is further achieved. This work offers an effective strategy for constructing UV-ultrasensitive smart sensor and artificial vision systems.

## Results

The artificial vision system is composed of a sensing unit, a memory unit and a processing unit (Fig. 1a), the photoreceptor cell is a specialized type of neuroepithelial cell in retina with the ability of converting light into electrical impulses. Inspired by retina, the BTBTT6-

syn-based OPTs (Fig. 1b) was proposed to perform the sensing function of PRC. BTBTT6-syn is a kind of BTBT-based asymmetric molecules (inset of Fig. 1c), with much denser packing bilayer-type alignment structures and stronger intermolecular interactions within the layers. Due to the unique structure, BTBTT6-syn thin-film owns the ability of effective charge transport[30,31]. Moreover, BTBTT6-syn thin-film exhibits strong and unique absorption in UV region with the centered peaks around 370 nm (Fig. 1c), indicating its promising application as active layer in UV-sensitive OPTs. As shown in Fig. 1d, the device array using $6 \times 8$ patterns were prepared with bottom-gate top-contact (BGTC) configuration. The responsive properties of OPTs with bare $SiO_2$ dielectric were first studied, and the typical transfer characteristics under 370 nm ultraviolet with different light intensity were shown in Fig. 1e. Positive photoresponse were observed with the obvious response discrimination at each illumination intensity. More importantly, the fabricated devices could even detect the UV light intensity as low as $31\,nW\,cm^{-2}$, which was the lowest detectable intensity in organic UV-sensitive phototransistors (Fig. 1f, Table S1)[29,32–43]. By calculating the P based on the ratio of photo-generated current to the dark current under a certain light intensity[44], the maximum value of P could reach $1.2 \times 10^4$ under 370 nm wavelength illumination at $1600\,nW\,cm^{-2}$ light intensity (Fig. S1a). In addition, the calculated R and D* according to the formula in previous reported literature were also shown in Fig. S1b and S1c, respectively[23].

An ultrahigh value of R closing to $10^7\,A\,W^{-1}$ was obtained with ultrahigh detectivity ~$2 \times 10^{17}$ Jones, which presented the strong ability of $SiO_2$-based devices to detect and convert ultraweak light into electric current. To further investigate the effect of dielectric interface on the device performance, three kinds of buffer layers were used to modify bare $SiO_2$, including polystyrene (PS), poly (methyl methacrylate) (PMMA) and polyphenylene oxide (PPO), respectively. It has been verified that there is a large density of hydroxyl groups in the form of silanols on the surface of $SiO_2$[45], the introduction of buffer layer could effectively reduce these electron-trapping sites. As shown in Fig. S2, with the aid of the modified layer, the off-state current decreased from $10^{-10}$ to $10^{-12}\,A$ and the on-state current accordingly increased. As a result, it was found that under the intensity of $1600\,nW\,cm^{-2}$, the value of P increased by at least one order of magnitude and the best value could reach to $8 \times 10^5$ (Fig. S1d and S3), which is the best result among the organic UV-sensitive phototransistors (Fig. 1g, Table S1)[29,32–43]. Although the enhancement of P value under $1600\,nW\,cm^{-2}$ illumination was observed, the devices with buffer layer showed a weak detection ability of ultraweak UV light compared with bare $SiO_2$ dielectric layer (Fig. 2a and S4). One possible explanation is that the buffer layer could reduce the trap density ($N_T$) on the bare $SiO_2$ interface (Fig. S5)[46,47]. In order to verify the trap dominant effect at ultraweak illumination intensity, the photocurrent with respect to the illumination intensity was plotted in Fig. 2b, c and S6. A linear increase of the photocurrent with the illumination intensity was observed at light intensity from 31 to $123\,nW\,cm^{-2}$ (Area I), indicating a photoconductive effect for photoresponse operating mechanism[22,46]. Under the light intensity above $123\,nW\,cm^{-2}$ (Area II), the nonlinearity determined that the photovoltaic mode had a decisive influence on the photoresponse operating process[24,48]. On this basis, when the device was illuminated under ultraweak UV light (Area I), free electrons and holes are generated in the light-absorbing BTBTT6-syn thin-film layer. Compared with the devices using buffer layers, a large number of photogenerated electrons was trapped by the large density of hydroxyl groups on $SiO_2$ surface, leading to the left free holes easily drifted toward the drain electrodes driven by the electric field for the realization of ultraweak light detection.

In addition, to perform comparative experiments, the phototransistors based on pure polymer single-component dielectric layers (PS, PMMA and PPO) on ITO/glass substrate without $SiO_2$ dielectric layer were also fabricated, which showed lower trap density than the

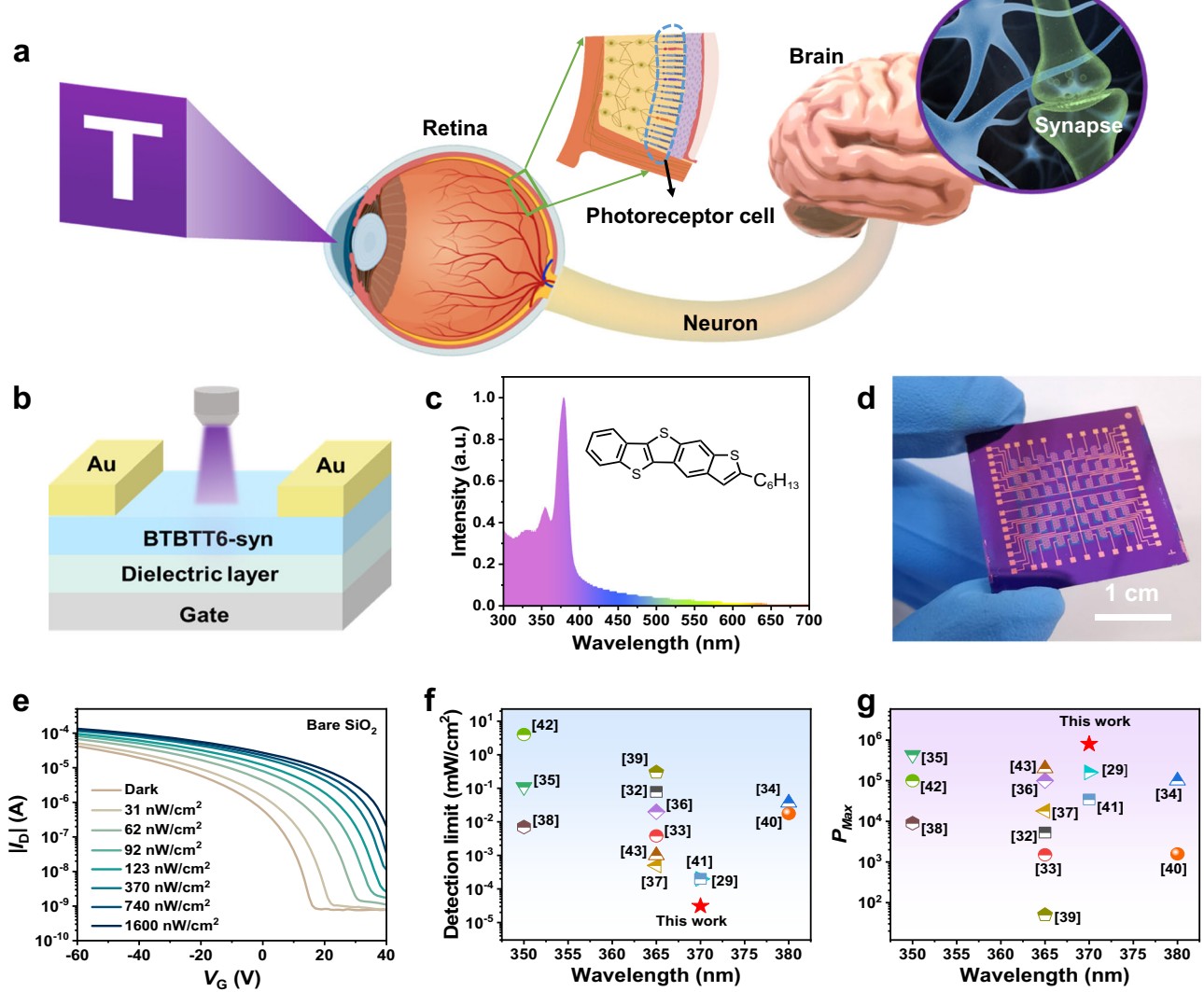

**Fig. 1 | The UV-ultrasensitive neuromorphic vision sensor that uses organic phototransistors. a** The artificial vision system composed of a sensing unit, a memory unit and a processing unit. **b** The schematics of neuromorphic vision organic sensor. **c** UV-vis absorption spectra for BTBTT6-syn; inset: chemical structure of BTBTT6-syn. **d** The 6 × 8 patterned device array. **e** The transfer curves of the OPTs based on bare $SiO_2$ dielectric in various illumination intensities. **f** The detection limit of our OPTs is compared with that previously reported, each symbol represents the corresponding reference. **g** $P_{max}$ of our OPTs is compared with that previously reported, each symbol represents the corresponding reference.

device with $SiO_2$ dielectric layer (Fig. S7) and the linear photoresponse occurred (Fig. S8), also resulting in the weak detection of ultraweak UV light (Fig. 2d and S9). Nevertheless, when the light intensity increased and the device was working at area II, the synergistic effect of the exciton binding energy and device carrier mobility also played an important role in modulating this optoelectrical process. As shown in Fig. 3a, the temperature-dependent photoluminescence (PL) spectra of BTBTT6-syn at bare $SiO_2$ interfaces was measured. The exciton binding energy was obtained by fitting the integral area at different temperatures with Arrhenius equation:[49]

$$I(T) = I_0/(1 + Ae^{-E_B/kT}) \qquad (1)$$

where $I_0$ is the intensity at 0 K, $k$ is the Boltzmann constant, and $T$ is the temperature. It was calculated that the $E_B$ on bare $SiO_2$ interface is 127.8 meV (Fig. 3b). Interestingly, the introduction of buffer layers could decrease the exciton binding energy with the value of $E_B$ based on $SiO_2/PS$, $SiO_2/PMMA$, and $SiO_2/PPO$ interfaces about 13.2 meV, 16.0 meV, 14.8 meV, respectively (Fig. 3c–f), indicating the smaller energy required for the separation of photogenerated excitons into

free electrons and holes on the buffer layer modified interfaces. In addition, the charge carrier transport properties of these devices were further investigated. It was found that the morphologies (Fig. S10) and molecular stacking (Fig. S11) of BTBTT6-syn thin-film were not changed that much on different surfaces. Therefore, the increase of device mobility using buffer layers (Fig. S12) could be mainly ascribed to the decrease of interface trap density (Fig. S5). As discussed above, the collective effect of the lower exciton binding energy and higher carrier mobility determined the increase of $P$ value with the aid of buffer layers under the illumination intensity from 370 to 1600 nW cm$^{-2}$. Meanwhile, the value of $R$ (Fig. S13a and S14) and $D^*$ (Fig. S13b and S15) was not correspondingly improved compared with $SiO_2$-based device, which can be explained by the relatively weak ability to detect and converting ultraweak light into electric current using buffer layers modified devices. It is worth mentioning that all the best value of $R$ above $10^6$ A W$^{-1}$ and $D^*$ above $10^{16}$ Jones can be achieved on different dielectric layers, and the performance of our OPTs in all aspects is compared with that reported in previous literature, as shown in Table S1, demonstrating the strong detection ability of weak light by using BTBTT6-syn-based phototransistors. Besides the detection to

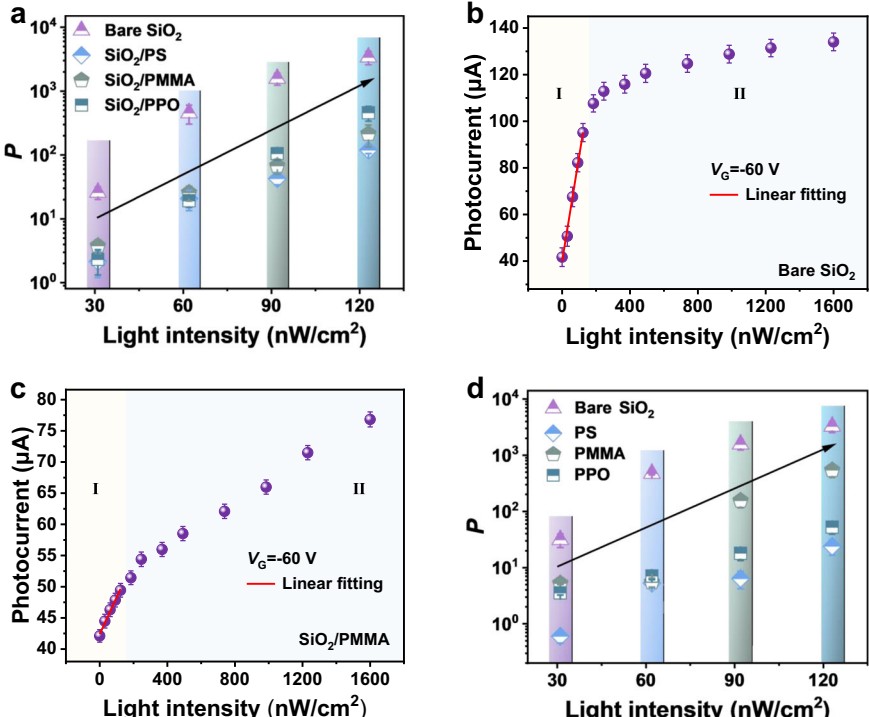

**Fig. 2 | The photosensitivity and photocurrent of OPTs under various illumination intensities. a** $P$ in different interface (bare and buffer layers modified SiO$_2$) as a function of low illumination intensity. **b** The dependence of photocurrent on light intensity based on bare SiO$_2$. **c** The dependence of photocurrent on light intensity based on bare SiO$_2$/PMMA. **d** $P$ of the phototransistor based on bare SiO$_2$ dielectric comparison with single-component PS, PMMA, PPO at low illumination intensity. The values and error bars in Fig. 2b and 2c were the mean value and standard deviation obtained with 10 OPT devices.

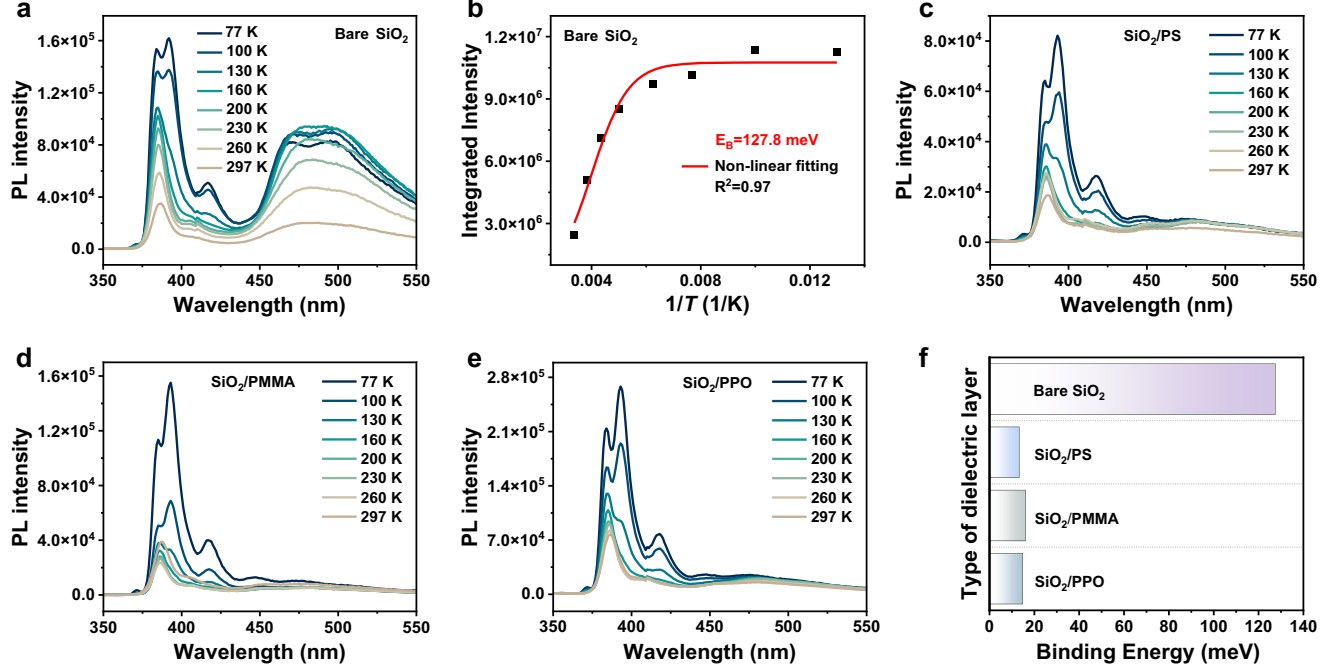

**Fig. 3 | Temperature-dependent photoluminescence (PL) spectra.** PL spectra for BTBTT6-syn films with **a** Bare SiO$_2$, **c** SiO$_2$/PS, **d** SiO$_2$/PMMA, and **e** SiO$_2$/PPO. **b** Temperature-dependent data of integrated intensity for BTBTT6-syn films with bare SiO$_2$ dielectric. **f** Binding energy with different interfaces.

weak light, we continue to study the capability of device adapting itself to UV light with high illumination intensity. It was found that the positive photoresponse was still observed from the device based on bare SiO$_2$ and PMMA/SiO$_2$ even the illumination intensity increased to 24.1 mW cm$^{-2}$ (Fig. S16a and 16b) possible due to the trap dominant effect (Fig. S5), but the maximum photocurrent has converged to the saturation state. Since there were less electron trap density at the interface of PS/SiO$_2$ and PPO/SiO$_2$ device (Fig. S5), and BTBTT6-syn could form relatively smaller hole barrier with PPO and PS (Fig. S17), photogenerated holes could be trapped in the high negative bias

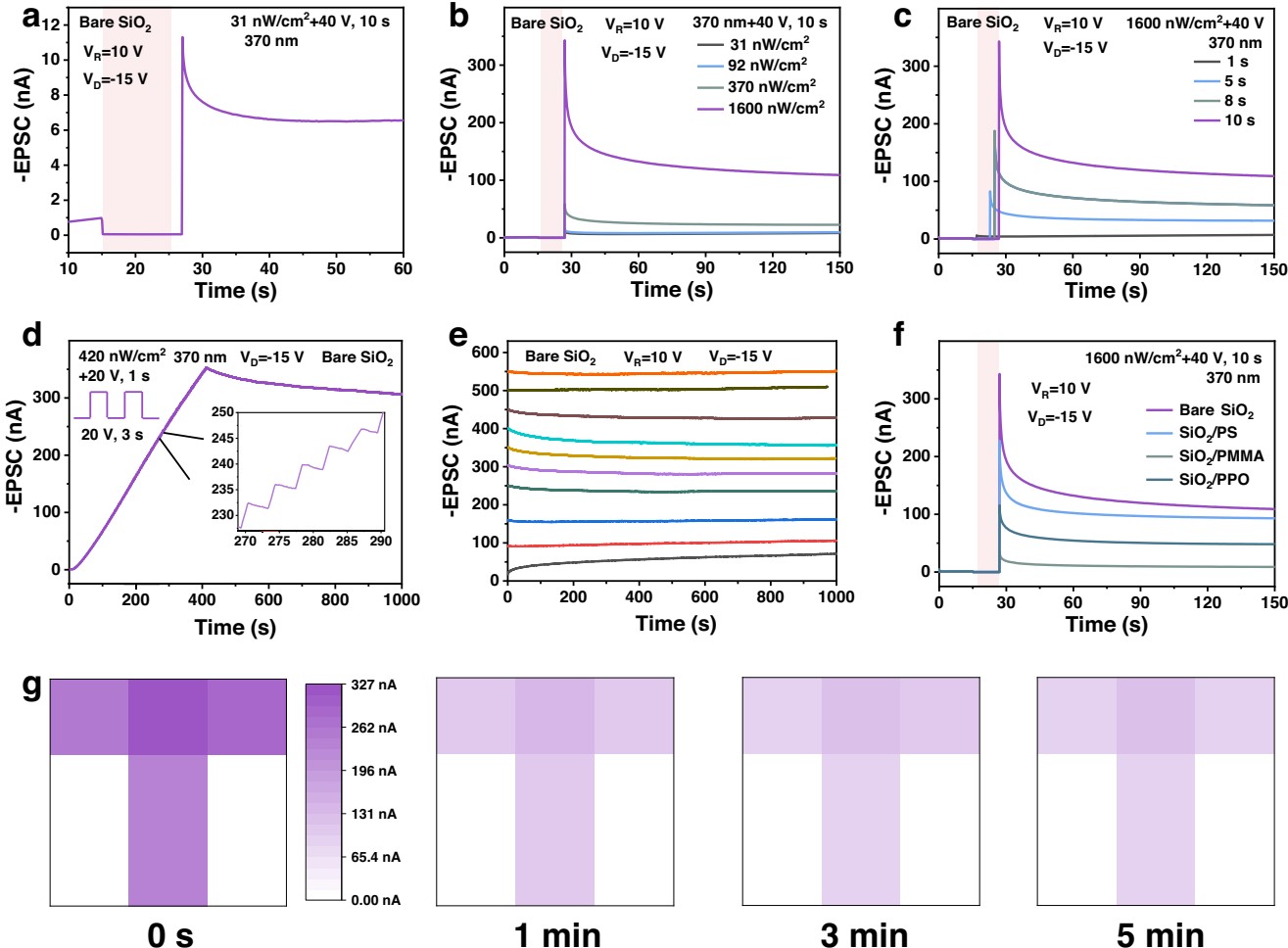

**Fig. 4 | Synaptic plasticity and image memorization.** ESPC of the bare SiO$_2$ device triggered by: **a** an ultraweak UV light spike; **b** various UV illumination intensities; **c** various pulse widths of UV light; **d** 100 consecutive UV presynaptic spikes. **e** The retention property of different current levels for the bare SiO$_2$ device. **f** EPSC of the devices with different buffer layers. **g** Image detection and memorization of the bare SiO$_2$ device array with an input image of letter T. All the tests were carried out under UV light stimulation. The light pink shading regions in **a**–**c** and **f** represent the duration of UV light stimuli. $V_D$ and $V_R$ in 4**a**–**f** represent for the drain voltage and reading gate voltage, respectively.

voltage region under high illumination intensity (e.g., 24.1 mW cm$^{-2}$), resulting in the decrease of the photocurrent (Fig. S16c and 16d). In addition, the above-mentioned two phenomena are also observed in the devices with single-component dielectric layer (e.g., PS, PPO, and PMMA) (Fig. S17).

Based on the ultraweak UV light detection and interface charge trapping ability, the synaptic OPTs were further investigated for emulating light information memory and processing of the visual perception system[7]. Figure 4a showed the typical synaptic plasticity of excitatory postsynaptic current (EPSC) of the bare SiO$_2$ device. In this case, light pulse and the drain current were regarded as the presynaptic input and postsynaptic signal, respectively. After the ultra-weak UV light spike (31 nW cm$^{-2}$, 10 s) with a $V_G$ bias of 40 V was applied, a sharp increase in EPSC can be observed with the peak current value of 11.3 nA. Then the EPSC gradually decayed to a stable current value of 6.5 nA in ~10 s. The stable PSC of 6.5 nA could be maintained, indicating the property of long-term depression (LTP) and endowing the device as a component with integrated memory and sensing functions, rather than a pure detector device[50]. Figure 4b, c showed that the synaptic strength of LTP could be enhanced by increasing the illumination intensity or the duration of light pulses, with the significantly increased peak value of the EPSC and thus the high dynamic range of weight change. All the EPSC curves are tested

after the appropriate negative voltage pulse to reset the device to the initial state of about 1 nA. The time response of the synaptic OPTs could be decreased to 100 ms under the illumination intensity of 1600 nW cm$^{-2}$ (Fig. S18). In addition, the LTP characteristic was also dependent on the number of light pulses. An increased number ($N = 100$) of presynaptic spikes was applied (Fig. 4d), and the EPSC increased linearly after the spikes, demonstrating linear weight update with optical writing. The performance of LTP was further verified by retention tests. The nonvolatile retention time of 1000 s based on 10 different current states were tested as shown in Fig. 4e, with the best nonvolatile retention time of 20,000 s achieved (Fig. S19). Besides, the EPSC of the devices with a different buffer layer (PS, PMMA, and PPO) were also measured (Fig. 4f and S20). Obviously, the dynamic range of weight change could be well modulated by the interfacial trapping of the buffer layer.

With the integration of optical detection and retention capabilities, the as-prepared synaptic OPTs become a potential candidate of NeuVS for image sensing and memorization[51]. As shown in Fig. 4g, nine devices consisting of 3 × 3-pixel array were used to detect and memorize the input image of letter T, five devices were stimulated by a light pulse of 1600 nW cm$^{-2}$ for 10 s with an $V_G$ bias of 40 V. The letter was written to the NeuVS array and then recorded by measuring the change of EPSC over time. After the light spike (0 s), the currents of the

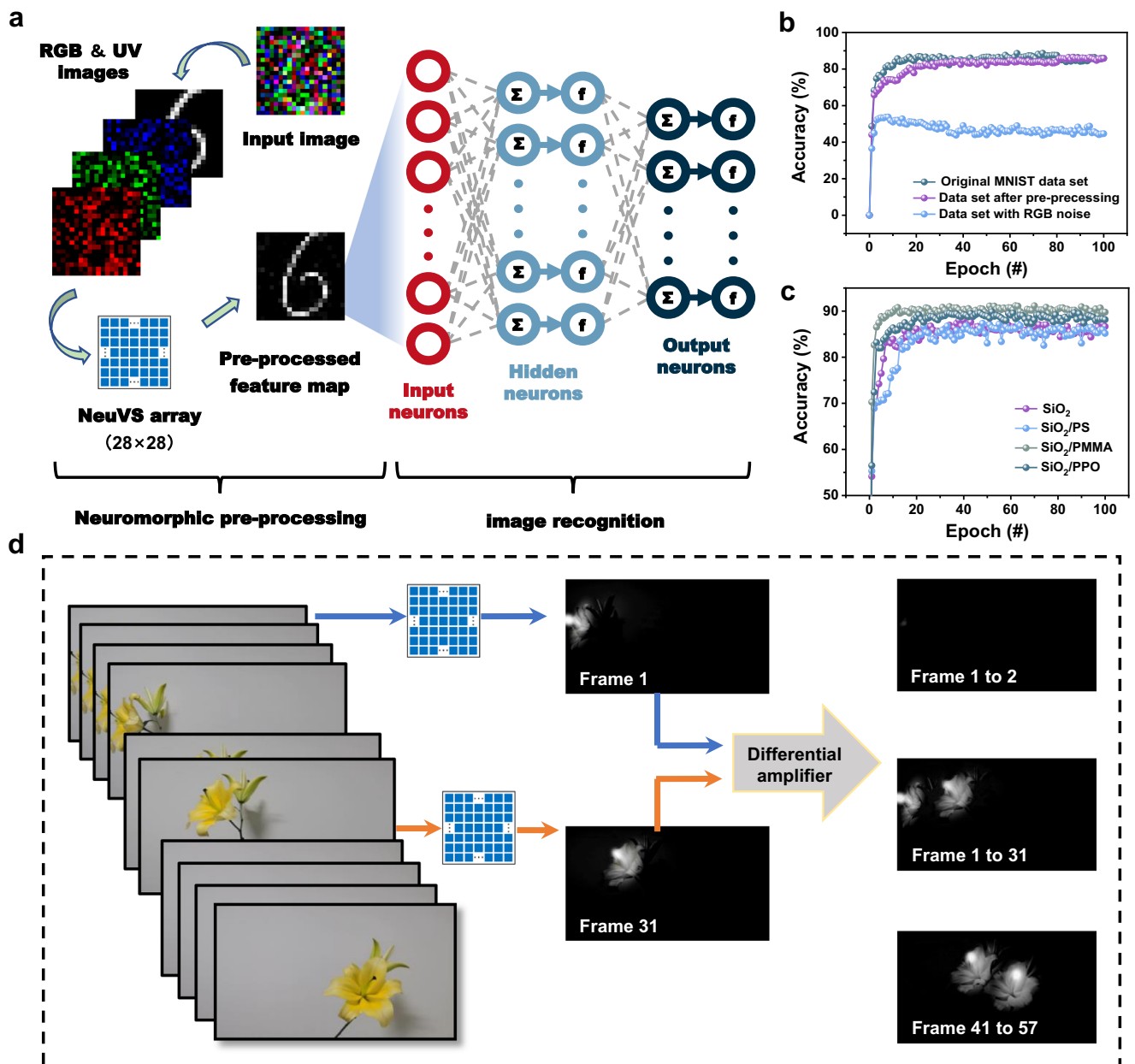

**Fig. 5 | Image recognition using the neuromorphic vision organic sensor. a** Schematic illustration of the pre-processing feature realization and the multilayer neural network for digit recognition. **b** Recognition accuracy with and without neuromorphic pre-processing. **c** Recognition accuracy with different buffer layers. **d** Illustration of motion detection with the BTBTT6-syn-based synaptic OPT arrays.

NeuVS array greatly increased. Then the currents decrease gradually (1 min to 5 min), and the device array performed a nonvolatile feature to UV image input. The devices with a different buffer layer (PS, PMMA, and PPO) were also characterized and shown in Fig. S21, which illustrated the BTBTT6-syn-based neuromorphic array owned good image perception and storage capability.

The EPSC of the bare $SiO_2$ device can also be triggered by RGB light spikes (650 nm, 520 nm, and 450 nm, Fig. S22 and S23), the EPSC and weight update values obtained under the illumination of RGB light were much lower than that of UV light, indicating the high UV selective response properties of the synaptic OPTs and the OPTs can be used to simulate the tetrachromatic vision. The vision system is fundamentally a sensing-memory-processing system, and the retina system can not only sense light, but also preform pre-processing and modification with amacrine cells and ganglions. For example, when focusing on the information given by the characteristic color, the

vision system can filter other colors. Benefiting from the UV-ultrasensitive and tunable synaptic plasticity, the OPT array can be used to simulate the pre-processing feature. As schematically shown in Fig. 5a, visual system was divided into two parts, neuromorphic pre-processing, and image recognition. In the former part, a convolution kernel was designed to simulate the extraction of UV information, the convolution kernel is a matrix of weight values used to perform a weighted average operation. A detailed explanation was described in Supplementary Note 1. And for the latter part, to demonstrate the advantage in image recognition with the pre-processing feature of focusing on UV information, a multilayer perceptron-based ANN (400 input neurons, 100 hidden neurons, and 10 output neurons) was simulated for image recognition[52].

During the image recognition, the Mixed National Institute of Standards and Technology (MNIST) handwritten image dataset was first used to train the ANN. Subsequently, three types of test datasets

including the original MNIST dataset, the dataset adding with RGB Gaussian noise, and the pre-processed dataset were input to the ANN for recognition. As shown in Fig. 5b, after 40 training epochs, the recognition accuracy was 46% for the dataset with RGB Gaussian noise, meaning that the recognition system can hardly recognize the characteristic information under conditions of RGB noise. By contrast, for the pre-processed dataset, the recognition accuracy improved to 84%, similar to that obtained for the original MNIST (85%), demonstrating the effectiveness of the OPT array in extracting ultraviolet information, and implying its potential application in real-time UV-monitoring systems. Moreover, as shown in Fig. 5c, for the devices with PS and PPO buffer layers, the corresponding recognition accuracy was 86 and 89%. The ANN based on the SiO₂/PMMA device can achieve a high recognition accuracy of 90% in favor of the high linearity in long-term potentiation and long-term depression (LTP/LTD) curves (Fig. S24), despite the low dynamic range of conductance margin.

Inspired by the principle that butterflies seek nectar, the UV-ultrasensitive characteristics of the BTBTT6-syn-based device enable us to demonstrate the motion detection of petals and pistils. (Fig. 5d and Supplementary Movie 1). After sensing with the device arrays, pistil signals in the memorized graphs are significantly enhanced, the petal signal is suppressed, and the surrounding signals are almost completely suppressed (Supplementary Movie 2). Subsequently, Supplementary Movie 2 is split into 74 frames with an interval of 20 ms, and the interframe differential computations are introduced to frame difference calculation of the memorized graphs[53]. The detailed explanation was described in Supplementary Note 2. By calculating the differences between the UV perceived and stored image brightness at $t$ and $t + \Delta t$ moment, for the objects without moving during $\Delta t$, the static pixels are suppressed to be almost zero. While for the moments with moving objects, the output image will contain the information about moving objects during the two moments. When the $\Delta t$ is one frame, the result of the motion detection can be seen in Supplementary Movie 3. The motion detection suggests that the proposed BTBTT6-syn-based devices can be used for both static and dynamic recognition.

## Discussion
By selecting BTBTT6-syn as the light-absorbing active layer, the high-performance UV-sensitive OPTs were prepared, which could detect ultraweak light as low as 31 nW cm⁻². The trap dominated in detection of ultraweak light was proposed and the synergistic effect of the exciton binding energy and carrier mobility was responsive for the higher light intensity detection. Taking advantage of interface engineering, the fabricated OPTs showed controllable modulation performance with the best value of $P$ closing $10^6$, $R$ as high as $10^7$ A W⁻¹ and $D^*$ above $10^{17}$ Jones. The optoelectronic synaptic functions with integrated sensing, nonvolatile multi-level memory, and neuromorphic computing features were successfully realized in the synaptic OPTs. The UV-ultrasensitive and tunable synaptic plasticity of the OPTs array enables us to perform static and dynamic image recognition by extract ultraviolet information from the color picture. The recognition accuracy was effectively improved to 90% by reducing redundant data. These results may provide a path to design UV-ultrasensitive OPTs for tetrachromatic vision sensors and provide the potential for the future integrated sensing-memory-processing for artificial vision systems.

## Methods
### Device fabrication
Bottom-gate top-contact OFETs were fabricated, SiO₂ was treated with O₂ plasma (50 W, 5 min), then the buffer layer was obtained by spin coating with a 10 mg/ml solution of the polymer (PS, PMMA, PPO) on SiO₂/Si wafer (SiO₂ 300 nm thick), and the single-component dielectrics were all obtained by spin-coating on the ITO with a 30 mg/ml solution for the preparation of bottom-gate top-contact OFETs, after

the buffer layers and single-component dielectrics are spin-coated, the samples are annealed at 80 °C for 30 min in a tubular furnace (BTF-1200C, Anhui Best Equipment Ltd., China). ITO glass substrates used in the study were successively cleaned with deionized water, acetone, pure ethanol, and then dried with nitrogen. The surface of ITO glass was treated with O₂ plasma (50 W, 5 min). Here, plasma treatment was carried out using Diener electronic GmbH+Co.KG (Germany). Then, in a vacuum of $1 \times 10^{-5}$ Pa, a 10 nm BTBTT6-syn film was deposited on the buffer layer or bare SiO₂ at a deposition rate of 0.04 Å s⁻¹; a metal mask was used to deposit 20 nm Au on the surface of the BTBTT6-syn at a deposition rate of 0.08 Å s⁻¹ as source/drain electrodes.

### Sample measurements
The phototransistors were characterized in the air by using a Keithley 4200-SCS analyzer and were illuminated by the assistance of the xenon lighted source as 370 nm UV ultraviolet. Commercial LED with the optical filter of 450 nm, 520 nm, and 650 nm and xenon lamp with a wavelength of 370 nm together with a Keithley 2636B semiconductor parameter analyzer was used during the measurement of synaptic characteristics. AFM images were recorded in tapping-mode utilizing a Bruker Dimension Icon microscope in atmospheric environment. Temperature-dependent photoluminescence spectra were characterized by FLS1000 spectrophotometer (Hitachi) from 77 to 297 K. The interfacial trap density ($N_T$) was estimated as a function of temperature ($T$) from the subthreshold swing ($S$), according to the following equation:

$$N_T = \frac{C_i}{e}\left(\frac{eS}{\ln 10 K_B T} - 1\right) \tag{2}$$

where $Ci$ denotes the capacitance per unit area across gate dielectric, and $e$ is the elementary charge, and charge mobility ($\mu$) was calculated according the following equation:

$$I_D = \left(\frac{W}{2L}\right)C_i\mu(V_G - V_T)^2 \tag{3}$$

## Data availability
The data related to the figures and other findings of this study are available from the corresponding author upon reasonable request.

## Code availability
The code used for the simulation is available from the corresponding author with detailed explanations upon reasonable request.

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

## Acknowledgements

This work was supported by the National Key Research and Development Program of China (2021YFA0717900), the National Natural Science Foundation of China (62004138, 52273190, 52121002, 21573277, 51503221, 61905121), Beijing National Laboratory for Molecular Sciences (BNLMS202006), the Natural Science Foundation of Jiangsu Province, China (no. BK20190734), the Haihe Laboratory of Sustainable Chemical Transformations and the Jiangsu Funding Program for Excellent Postdoctoral Talent (2022ZB402).

## Author contributions

D.J., H.L., and W.H. conceived and designed the experiments. H.T., Y.D., and Y.G. designed and synthesized organic semiconductor materials. T.J., Y.W., Y.Z., L.W., and X.H. performed the experiments. D.J., H.L., H.T., L. Y., H.D., L.L., L.X., and W.H. discussed the results. D.J., H.L., H.T., T.J., and Y.W. co-wrote the manuscript. All the authors read and commented on the manuscript.

## Competing interests

The authors declare no competing interests.
