## [Peer Review File · Nature Communications]

Reviewers' comments:

Reviewer #1 (Remarks to the Author):

The authors present an interesting work on a 2-D array for sensing weak UV signals using organic phototransistors. The authors examined several different imaging arrays based on the BTBTT6-syn organic phototransistors fabricated using bottom-gate top-contact configuration. The surface of the silicon dioxide was modified using polystyrene, poly methyl methacrylate and polyphenylene oxide to vary the electron trapping sites and its effects on UV sensitivity and linearity. Detailed measurements of the fabricated device are described in the paper validating the performance of their device to sense UV photons.

There are several major concerns regarding the manuscript that preclude this paper from publishing in Nature Communication.

1. One of the major claims of this paper is the highest UV sensitivity published to date. The authors claim that their device has UV sensitivity (detection limit) of 35nW/cm^2 at 370nm. However, the sensitivity of the device published in reference 27 is over two orders of magnitude higher: 0.2nW/cm^2 at 370 nm (see table S4 in ref 27). As a follow up on this claim, the detectivity of this device is lower than the device in ref. 27 (10^{17} vs. 10^{18} J). Hence, this claim in the paper is misleading.

2. One of the important aspects of a sensing element/device is photo response over time to different intensities. For example, ref. 27 in figure S17 presents photoresponse to a time varying stimulus. This kind of measurement is not provided in this paper. Based on the trapping mechanisms that is used in this device, dynamic photo response is probably not possible with this device, which is a major limitation in a photo sensitive device. The authors try to use this to their advantage to memorize an image in the device, though the retention time is limited to under an hour. In either case, both of these limitations should be clearly articulated in the paper.

3. The application data presented in Figure 5 is based on emulation of their device to recognize numbers and track flowers. Since this is an emulation data based on a computer model of their device, it has very limited contributions to the device capabilities. The authors achieve 90% detection accuracy based on the simulation of their device. However, today's state of the art machine learning algorithms perform over 99% detection accuracy on the same data base.

Since the resolution of the images used in the neural network is on par if they used an array of 3 by 3 of their imaging array, they should perform this experiments on real data recorded in the UV and/or visible spectrum with their device with a tailored neural network. If they authors use color information for detection, this should be carefully evaluated versus an approach that only uses UV data.

4. The title and introduction of the paper talk about tetrachromatic vision. I think this is misleading because the authors demonstrate imaging of only one wavelength/spectrum at a time. A tetrachromatic image sensor should be able to capture 4 different spectrums simultaneously, which is not demonstrated in this paper.

Reviewer #2 (Remarks to the Author):

In the manuscript the authors present an organic phototransistor (OPT) device that acts as a neuromorphic vision sensor (NeuVS) for UV light. The authors propose the NeuVS as a means to incorporate UV sensing into artificial vision systems that operate similar to examples of tetrachromat vision in nature. Initially, the authors introduce their OPT device and present numerous measurements showing the device's excellent performance. The device is compared to multiple reports of UV sensors in literature and the authors demonstrate that their design offers best in class characteristics. Importantly, the authors provide further in depth analysis of alternative device structures and analyse in great detail the incorporation of different buffer layers. This analysis reveals 2 potential mechanisms which allow device design to be targeted for more efficient ultraweak UV detection or for higher carrier mobility. The authors then go on to demonstrate the OPT devices being used as synaptic links that perform pre-processing as in a visual perception system. It is demonstrated that the synaptic devices are capable of retaining a stable current in memory and that this current can be controllably tuned using characteristics of the incoming light. Further, as in neuromorphic models, the OPT synapse can demonstrate the integration of optical signals given a high enough pulse rate, an interesting feature for potential processing applications. Finally, the authors provide two experimental demonstrations of pre-processing applications with their OPT devices. In the first demonstration UV filtering is applied in the context of a benchmark classification task, where the authors report an improvement in performance before and after OPT UV filtering. In the second demonstration the OPT device is employed as a dynamic vision sensor where UV detection is used to perceive a moving flower with improved clarity.

Overall, I believe the manuscript represents an interesting and significant body of work which has been carried out with a sound level of methodology. The results are of interest to the scientific community and a good level of data is provided in support of their findings. The research reported in the manuscript is of high interest to the fields of neuromorphic photonics and UV sensing, aligning

well with excitement growing in photonic based neuromorphic systems. The manuscript is written in a logical structure with good progression starting from the initial discussion of the device, to device development considerations, and finally through to device implementation. I believe the manuscript should be accepted for publication following improvements to the manuscript.

I have the following suggestions and questions for the authors:

1. Significant improvements to the quality of the writing language need to be made before publication. Throughout the manuscript I was at some points confused and disappointed by the writing of otherwise good scientific results.
2. In Fig. 3 all plots are missing Y-axis labels. The units are arbitrary however it makes comparisons difficult. I would like to see axis labels (similar to that of Fig. 1c) in Fig. 3 as well as others in the supplementary information.
3. In Fig. 3b a linear fitting is given, however unlike other plots throughout the paper it is more than a straight line fit. If a linear model is used to fit the data, it should be made clear that it is the case and the model should be provided.
4. The authors should make clear what device structures were used to reach stated performances and to carry out experiments. Given the detailed discussion of multiple structures it would be helpful to remind the reader.
5. Why is the initial EPSC in Fig 4a (and others in the supplementary information) non-zero? Further, the value then hits zero rapidly, is there a method to erase the stable EPSC for the rewriting of synaptic weight?
6. In the MNIST recognition the SiO₂/PMMA structure reached 90% accuracy, how would this value change if weak UV detection was used? Do you expect to see bare SiO₂ outperform the other structures?
7. I would appreciate the authors commenting on which device structure they endorse. Further do you see benefit in creating a two-in-one device structure for both weak UV detection and high light intensity benefits.
8. LTP is defined as both long-term potentiation and long-term plasticity. Further, LTD is mentioned on line 268 without previous definition or explanation.

Comment (No change required) – As a colourblind person I was struggling with some figures in the manuscript and the Supplementary information (greens and reds primarily). Small accessibility considerations make all difference and are very much appreciated.

Reviewer #3 (Remarks to the Author):

Mimicking tetrachromatic vision using neuromorphic sensors with ultraweak ultraviolet detection by Ting Jiang et al..

Nat. Comm 402683_0_

The authors demonstrate a neuromorphic vision sensor based on organic phototransistors. The work is interesting because it not only shows the photo-electrical characteristics of single devices but also the vision performance of a completely vision system inspired by signal pathways explored in nature.

The manuscript is well written and the supplementary part comprise a detailed summary of the photo transistor.

In Fig. 1(f) and Fig. 1(g) only the parameters detection limit and photosensitivity are compared to other publications. Here I propose to include a table comparing the following parameters with the literature as shown in a table:

Here an example from “Organic UV-Sensitive Phototransistors Based on Distriphenylamineethynylpyrene Derivatives with Ultra-High Detectivity”

Approaching 10^{18} , Advanced Materials, 2020, by Dan Liu et al..

DOI: 10.1002/adma.201907791, see page 1907791 (7 of 9), table 1.

In addition, in the table the parameters: Bias voltage, rise time and decay time. Such parameters should be compared between the performance of the device performance of this work to those best values published in the literature. In addition, the reference list should include the paper by Liu et al..

On page 2, the authors write (see line 62/63):

“Despite the emerging importance of tetrachromatic vision in advanced artificial visual system, relatively little attention has been paid to sensing and processing of UV light.”

In case there is a “little attention”, this works should be cited.

Response to Reviewer #1

We would like to thank the reviewer for his/her encouraging and valuable suggestions of our work and the helpful comments for further improvements. Please find below our responses (in blue) to each of your specific comments (in black). Revisions to the original manuscript are underlined in red.

Comment 1: One of the major claims of this paper is the highest UV sensitivity published to date. The authors claim that their device has UV sensitivity (detection limit) of 35 nW/cm² at 370nm. However, the sensitivity of the device published in reference 27 is over two orders of magnitude higher: 0.2 nW/cm² at 370 nm (see table S4 in ref 27). As a follow up on this claim, the detectivity of this device is lower than the device in ref. 27 (10¹⁷ vs. 10¹⁸ J). Hence, this claim in the paper is misleading.

Our reply: We appreciate the reviewer for the valuable comments. We carefully checked the Table S4 in Ref. 27, and found that there should be a misunderstanding. The minimum light intensity used in Ref. 27 is 0.2 μW cm⁻², but not 0.2 nW cm⁻² (as shown in the following figure). Moreover, the reviewer mentioned that the detectivity obtained in our work is lower than the device in ref. 27 (10¹⁷ J vs. 10¹⁸ J). We believe that the slightly higher detectivity is due to several factors, e.g., the channel material is single crystal rather than our thin film device, and the channel area is smaller. Moreover, the detectivity should be calculated based on a certain light intensity, since the detectivity in our work and ref. 27 are calculated with different light intensity, it might be inappropriate to make a comparison. Therefore, in our manuscript, our detected optical figures of merit including *P*, *R* and *D*^{*} presented **one of the highest values** among all organic photodetectors.

Table S4. Single crystal phototransistor performances of 1,6-DTEP and 2,7-DTEP.

Compound	Light source	P_i (mW cm ⁻²)	P	R (A W ⁻¹)	D [*] (Jones)
1,6-DTEP	white light	0.0345	7.30 × 10 ⁵	3.67 × 10 ⁷	4.21 × 10 ¹⁶
	white light	0.0790	8.84 × 10 ⁵	1.76 × 10 ⁸	2.23 × 10 ¹⁶
	white light	0.1580	9.74 × 10 ⁵	9.09 × 10 ⁷	1.23 × 10 ¹⁶
	370nm UV light	0.0020	8.68 × 10 ⁵	7.78 × 10 ⁵	8.85 × 10 ¹⁷
	370nm UV light	0.0002	1.60 × 10 ⁵	2.86 × 10 ⁶	1.49 × 10 ¹⁸
2,7-DTEP	white light	0.0345	1.40 × 10 ³	4.83 × 10 ⁷	1.14 × 10 ¹⁴
	white light	0.0790	2.74 × 10 ³	2.83 × 10 ⁷	9.74 × 10 ¹³
	white light	0.1580	3.60 × 10 ³	1.59 × 10 ⁷	6.41 × 10 ¹³
	370nm UV light	0.0020	4.70 × 10 ³	1.64 × 10 ⁸	7.28 × 10 ¹⁵
	370nm UV light	0.0002	4.35 × 10 ³	1.04 × 10 ⁸	5.28 × 10 ¹⁶

Comment 2: One of the important aspects of a sensing element/device is photo response over time to different intensities. For example, ref. 27 in figure S17 presents photoresponse to a time varying stimulus. This kind of measurement is not provided in this paper. Based on the trapping mechanisms that is used in this device, dynamic photo response is probably not possible with this device, which is a major limitation in a photo sensitive device. The authors try to use this to their advantage to memorize an image in the device, though the retention time is limited to under an hour. In either case, both of these limitations should be clearly articulated in the paper.

Our reply: The comments and suggestions of the reviewer are very valuable. In fact, our devices were artificial synaptic transistors with integrated memory and sensing functions rather than pure switching (detector) devices. Therefore, we cannot do the switching measurements similar to the Figure S17 in Ref. 27. According to your suggestions, the corresponding limitation has been added in the revised manuscript. Moreover, we have tested the retention property of the device and given in **Figure S19**, and the sentence “with the best nonvolatile retention time of 20000 s achieved (**Figure S19**)” was expressed in the manuscript. The programmed state can stay stable even after 20000 s, but not limited to under an hour.

Our revision: The stable PSC of 6.5 nA could be maintained, indicating the property of long-term plasticity (LTP) and endowing the device as a component with integrated memory and sensing functions, rather than a pure detector device⁵⁰.

Comment 3: The application data presented in Figure 5 is based on emulation of their device to recognize numbers and track flowers. Since this is an emulation data based on a computer model of their device, it has very limited contributions to the device capabilities. The authors achieve 90% detection accuracy based on the simulation of their device. However, today's state of the art machine learning algorithms perform over 99% detection accuracy on the same data base.

Since the resolution of the images used in the neural network is on par if they used an array of 3 by 3 of their imaging array, they should perform this experiments on real data recorded in the UV and/or visible spectrum with their device with a tailored neural network. If they authors use color information for detection, this should be carefully evaluated versus an approach that only uses UV data.

Our reply: Thank you for your valuable comments. We understand that the reviewer’s concern about the advance of our work. We do agree with the reviewer 1 that today’s state of the art machine learning algorithms perform over 99% detection accuracy. Actually, such a high level of accuracy is achieved by pure algorithms. However, the 90% detection accuracy is totally determined by our device performance, not a complete algorithmic process. More importantly, as exhibited in **Figure 5**, the neuromorphic vision sensor (NeuVS) array not only performs as the signal pre-processing system but also as the testing dataset. Benefit from the pre-processing function, the recognition accuracy is significantly improved and similar to that obtained for the original MNIST dataset. The following list shows the recognition accuracy of recent organic synaptic transistors. It can be seen that the recognition accuracy obtained in our work is competitive to the reported works.

Device structure	Recognition accuracy (%)	Ref.
Au/PDPP4T/chlorophyll-a/WCN/Au	79	1
Au/Pentacene/PMMA/CsPbBr ₃ QDs/ Al ₂ O ₃ /Au/PI	~75	2
Au/PDVT-10/DAE/SiO ₂ /Si	85	3
Au/PDPP4T/chlorophyll/SiO ₂ /Si	82	4
Au/Pentacene/P(VDF-TrFE)/Ag	90	5
Au/CuPc/p-6P/SiO ₂ /Si	87.11	6
Au/PDVT-10/CdSe/ZnS/Al ₂ O ₃ /P(VDF-TrFE)/Si	95.7	7
Au/PDVT-10/P(VDF-TrFE)/Si	91.38	8
Au/DPP-DTT/PVDF-HFP/Al/PI	90.4	9

Cu/Pentacene/PMMA/ Zr-CsPbI ₃ /SiO ₂ /Si	76.2	10
Au/BTBTT6-syn/PMMA/ SiO ₂ /Si	90	This work

1. Zhang, J.Y. et al. Bioinspired organic optoelectronic synaptic transistors based on cellulose nanopaper and natural chlorophyll-a for neuromorphic systems. *npj Flexible Electronics*. **30**, 1888 (2022).
2. Li, M.Z. et al. Inorganic perovskite quantum dot-based strain sensors for data storage and in-sensor computing. *ACS Appl. Mater. Interfaces*. **13**, 30861-30873 (2021).
3. Li, E.L. et al. High-density reconfigurable synaptic transistors targeting a minimalist neural network. *ACS Appl. Mater. Interfaces*. **13**, 28564–28573 (2021).
4. Yang, B. et al. Bioinspired multifunctional organic transistors based on natural chlorophyll/organic semiconductors. *Adv. Mater.* **32**, 2001227 (2020).
5. Ham, S. et al. One-dimensional organic artificial multi-synapses enabling electronic textile neural network for wearable neuromorphic applications. *Sci. Adv.* **6**, eaba1178 (2020).
6. Qian, C. et al. Solar-stimulated optoelectronic synapse based on organic heterojunction with linearly potentiated synaptic weight for neuromorphic computing. *Nano Energy*. **66**, 104095 (2019).
7. Yu, R.J. et al. Programmable ferroelectric bionic vision hardware with selective attention for high-precision image classification. *Nat Commun.* **13**, 7019 (2022).
8. Li, E.L. et al. Nanoscale channel organic ferroelectric synaptic transistor array for high recognition accuracy neuromorphic computing. *Nano Energy*. **85**, 106010 (2021).
9. Yu, R.J. et al. Artificial tactile recognition enabled by flexible low-voltage organic transistors and low-power synaptic electronics. *ACS Appl. Mater. Interfaces*. **14**, 48948–48959 (2022).
10. Shao, H. et al. A reconfigurable optoelectronic synaptic transistor with stable Zr-CsPbI₃ nanocrystals for visuomorphic computing. *Adv. Mater.* 2208497 (2023).

Comment 4: The title and introduction of the paper talk about tetrachromatic vision. I think this is misleading because the authors demonstrate imaging of only one wavelength/spectrum at a time. A tetrachromatic image sensor should be able to capture 4 different spectrums simultaneously, which is

not demonstrated in this paper.

Our reply: Thank you for your valuable comments. We think there might be a misunderstanding of the starting point and purpose of our discussion of tetrachromatic vision. Since the human eyes cannot see UV light, which is harmful to human eyes, UV light detection is particularly important. Inspired by the tetrachromatic vision in biological vision system, we use neuromorphic devices to detect UV light, and hence realize the recognition and memorization of the static and dynamic images under UV light. In fact, our device can respond to both red, green and blue light (**Figure S22**), but they are more responsive to UV light. These are what we really focused on, and the starting point and purpose are clearly expressed in the introduction section. To avoid the misunderstanding, we have modified the title and the corresponding expressions in abstract section.

Our revision:

Title: Tetrachromatic vision-inspired neuromorphic sensors with ultraweak ultraviolet detection

Abstract section: Sensing and recognizing invisible ultraviolet (UV) light is vital for exploiting advanced artificial visual perception system. However, due to the uncertainty of the natural environment, the UV signal is very hard to be detected and perceived. Here, inspired by the tetrachromatic visual system, we report a controllable UV-ultrasensitive neuromorphic vision sensor (NeuVS) that uses organic phototransistors (OPTs) as the working unit to integrate sensing, memory and processing functions.

Response to Reviewer #2

We would like to thank the reviewer for his/her encouraging and valuable suggestions of our work and the helpful comments for further improvements. Please find below our responses (*in blue italic*) to each of your specific comments. Revisions to the original manuscript are underlined in red.

Comment 1: Significant improvements to the quality of the writing language need to be made before publication. Throughout the manuscript I was at some points confused and disappointed by the writing of otherwise good scientific results.

Our reply: *Thank you for your kind suggestion. We have carefully checked the language in the manuscript and the corresponding modification can be seen in the revised manuscript.*

Comment 2: In Fig. 3 all plots are missing Y-axis labels. The units are arbitrary however it makes comparisons difficult. I would like to see axis labels (similar to that of Fig. 1c) in Fig. 3 as well as others in the supplementary information.

Our reply: *Thank you for your valuable comments. The corresponding axis labels have been added in the manuscript and the supplementary information.*

New Figures:

Fig. 3 Temperature dependent photoluminescence (PL) spectra. PL spectra for BTBTT6-syn films with **a** Bare SiO₂, **c** SiO₂/PS, **d** SiO₂/PMMA and **e** SiO₂/PPO. **b** Temperature dependent data of integrated intensity for BTBTT6-syn films with bare SiO₂ dielectric. **f** Binding energy with different interfaces.

Figure S5. Trap density with bare SiO₂, SiO₂/PS, SiO₂/PMMA, SiO₂/PPO.

Figure S7. Trap density with bare SiO₂, PS, PMMA, PPO.

Figure S12. Mobility with bare SiO₂, SiO₂/PS, SiO₂/PMMA, SiO₂/PPO.

Comment 3: In Fig. 3b a linear fitting is given, however unlike other plots throughout the paper it is more than a straight line fit. If a linear model is used to fit the data, it should be made clear that it is the case and the model should be provided.

Our reply: *We are sorry for the typo in Fig. 3b, in which the fit should be non-linear fitting, and*

the correction has been made in the revised manuscript. The exciton binding energy is obtained by fitting the integral area at different temperatures with Arrhenius equation:

$$I(T) = I_0 / (1 + Ae^{-E_B/kT})$$

New Figures

Fig. 3 Temperature dependent photoluminescence (PL) spectra. PL spectra for BTBTT6-syn films with a Bare SiO₂, c SiO₂/PS, d SiO₂/PMMA and e SiO₂/PPO. b Temperature dependent data of integrated intensity for

BTBTT6-syn films with bare SiO₂ dielectric. **f** Binding energy with different interfaces.

Comment 4: The authors should make clear what device structures were used to reach stated performances and to carry out experiments. Given the detailed discussion of multiple structures it would be helpful to remind the reader.

Our reply: *Thank you for your valuable comments. Based on the different dielectric layers, there are three types of devices:*

1) The most basic type is the device based on bare SiO₂ dielectric layer with large density of hydroxyl groups in the form of silanols on the surface of SiO₂, the bare SiO₂ device owns good UV detection and synaptic performance for ultraweak light.

2) To further investigate the role of trap density in light detection, we introduced the devices with different buffer layer on SiO₂. The imported buffer layer reduces the surface trap density, improves the carrier mobility and reduces the exciton binding energy, so that the photosensitivity of the devices is further improved under high UV illumination.

3) In order to perform comparative experiments to verify the above conclusions, we fabricated the single-component pure polymer dielectric layer devices on ITO glass without SiO₂ dielectric layer.

To remind the reader to better understand the corresponding section, we have made a more detailed explanation of the single component phototransistors in the revised manuscript.

Our revision:

In addition, to perform comparative experiments, the phototransistors based on pure polymer single component dielectric layers (PS, PMMA and PPO) on ITO/glass substrate without SiO₂ dielectric layer were also fabricated.

Comment 5: Why is the initial EPSC in Fig 4a (and others in the supplementary information) non-zero? Further, the value then hits zero rapidly, is there a method to erase the stable EPSC for the rewriting of synaptic weight?

Our reply: *In this work, all the EPSC curves are tested after the appropriate negative voltage pulse to reset the device to the initial state of about 1 nA. However, in a relatively larger range of Y-axis labels (such as the range of -25 nA to 400 nA in Fig 4b), the initial state of EPSC seems to be “zero”, while with a relatively smaller range of Y-axis labels (such as the range of -1 nA to 13 nA in*

Fig 4a), the initial state seems to be “non-zero”. In Fig 4a, during the time range of 15 s to 27 s, UV-light spike (duration 10 s) with a V_G bias of 40 V was applied, thus, the of reading I_D is obtained with the reading V_G of 40 V, and the absolute value (that is, the -EPSC) is about 5×10^{-11} A, which seems to be “the value then hits zero rapidly”.

For clarity, the relevant description is given in the revised manuscript and the stimulation process is labeled in the corresponding figure. Thank you for your kind reminder.

Our revision:

Figure 4b and 4c showed that the synaptic strength of LTP could be enhanced through increasing the illumination intensity or the duration of light pulses, with the significantly increased peak value of the EPSC and thus the high dynamic range of weight change. All the EPSC curves are tested after the appropriate negative voltage pulse to reset the device to the initial state of about 1 nA.

Comment 6: In the MNIST recognition the SiO₂/PMMA structure reached 90% accuracy, how would this value change if weak UV detection was used? Do you expect to see bare SiO₂ outperform the other structures?

Our reply: *Thank you for your valuable comments. The recognition accuracy is affected by several parameters, such as the nonlinearity, symmetry and maximum/minimum ratio (G_{max}/G_{min}) of channel conductance, et al (please refer to our review paper for more details: Applied Physics Reviews, 2020, 7, 011307; Journal of Materials Chemistry C, 2021, 9, 11464).*

Generally, the G_{max}/G_{min} would become larger with increasing the UV light intensity, and thus increasing the accuracy. So, for the SiO₂/PMMA device, the recognition accuracy may become slightly better with higher UV light.

Due to the large density of hydroxyl groups in the form of silanols on the surface of SiO₂, it is easier for bare SiO₂ device to trap electrons and form a stable storage. As a result, it is difficult for bare SiO₂ device to obtain good LTP/LTD symmetry. Therefore, the recognition accuracy is difficult to be better than SiO₂/PMMA device.

Comment 7: I would appreciate the authors commenting on which device structure they endorse. Further do you see benefit in creating a two-in-one device structure for both weak UV detection and high light intensity benefits.

Our reply: Thank you for your valuable comments. In this work, we focused on the ultraweak UV-light detection, and hence realize the recognition and memorization of the static and dynamic images under weak UV light. In this regard, the bare SiO₂ device may perform best compared to other devices due to its ultraweak UV-light detection and good charge trapping ability. Benefiting from the ultraweak UV detection and the high light tolerance, our device has a wide light intensity detection window, which will have potential applications in dynamically adaptive vision sensors.

Comment 8: LTP is defined as both long-term potentiation and long-term plasticity. Further, LTD is mentioned on line 268 without previous definition or explanation.

Our reply: We apologize for the resulting confusion. To response, LTP is uniformly defined as long-term potentiation in the revised manuscript. Besides, we didn't use LTD independently in the manuscript. LTP/LTD is defined as long-term potentiation and long-term depression.

Comment (No change required) – As a colorblind person I was struggling with some figures in the manuscript and the Supplementary information (greens and reds primarily). Small accessibility considerations make all difference and are very much appreciated.

Our reply: We feel sorry that we didn't take the above into account when producing the figures, we have changed all the figures with the red and green colors together in the manuscript and the supplementary information, except for the Fig. S22 and Fig. S23, in which the red and green represent light colors. We have made corresponding text description in Fig. S22 and Fig. S23.

New Figures with corresponding text description:

Figure S22. EPSC of bare SiO₂ device triggered by various wavelengths illumination **a** RGB; **b** RGB and UV.

Figure S23. LTP of bare SiO₂ device triggered by various **a** wavelengths; **b** illumination intensities.

Response to Reviewer #3

We would like to thank the reviewer for his/her encouraging and valuable suggestions of our work and the helpful comments for further improvements. Please find below our responses (in blue) to each of your specific comments (in black). Revisions to the original manuscript are underlined in red.

Comment 1: In Fig. 1(f) and Fig. 1(g) only the parameters detection limit and photosensitivity are compared to other publications. Here I propose to include a table comparing the following parameters with the literature as shown in a table: Here an example from “Organic UV-Sensitive Phototransistors Based on Distriphenylamineethynylpyrene Derivatives with Ultra-High Detectivity” Approaching 10^{18} , *Advanced Materials*, 2020, by Dan Liu et al.. DOI: 10.1002/adma.201907791, see page 1907791 (7 of 9), table 1.

In addition, in the table the parameters: Bias voltage, rise time and decay time. Such parameters should be compared between the performance of the device performance of this work to those best values published in the literature. In addition, the reference list should include the paper by Liu et al.

Our reply: Thank you for your kind suggestions. According to your suggestions, we have included **Table S1** in the revised supplementary information. Moreover, we further added the parameters of bias voltage, rise time and decay time in **Table S1**. Due to the fact that our devices were artificial synaptic device, but not switching device, we cannot to provide rise time and decay time. We have explained **Table S1** in more detail in the manuscript.

Our revision:

.....which was the lowest detectable intensity in organic UV-sensitive phototransistors (Fig. 1f, Table S1)which is the best result among the organic UV-sensitive phototransistors (Fig. 1g, Table S1)

New table:

Table S1. Comparison of current work with representative phototransistors.

Response material	Detection limit (mw cm ⁻²)	Wave length (nm)	V _D (V)	Rise time	Decay time	P _{Max}	R _{Max} (A W ⁻¹)	D* _{Max} (Jones)	Growth mode	Ref.
DFD+TMT ES-P	0.078	365	-60	N/A	N/A	5.3 × 10 ³	5.3 × 10 ³	2.78 × 10 ¹⁴	cocrystal	1
C8-BTBT+Ag BiS ₂ QD	0.0038	365	-40	0.23 s	0.82 s	1.5 × 10 ³	20	4 × 10 ¹³	hybrid films (spin-coating)	2
BBDTE	0.037	380	-30	N/A	N/A	10 ⁵	9821	N/A	single crystal	3
BBDTY			-30	N/A	N/A	4429	9336	N/A	thin films	
BOPAnt	0.11	350	-60	N/A	N/A	4.34 × 10 ⁵	3100	N/A	single crystal	4
C8-BTBT + PLA	0.02	365	-30	40 s	80 s	1 × 10 ⁵	56	N/A	films (spin-coating)	5
DTTQ+ CsPbBr ₃ QD	0.0005	365	60	0.114 s	0.118 s	1.8 × 10 ⁴	7.1 × 10 ⁵	3.6 × 10 ¹³	films (solution-shearing)	6
1,6-DTEP	0.0002	370	-60	N/A	N/A	1.60 × 10 ⁵	2.86 × 10 ⁶	1.49 × 10 ¹⁸	single crystal	7
2,7-DTEP			-60	N/A	N/A	4.35 × 10 ³	1.04 × 10 ⁵	5.28 × 10 ¹⁶		
PC ₆₁ BM	0.007	315-400	80	0.94 ms	1.02 ms	9 × 10 ³	3 × 10 ³	1.3 × 10 ¹³	films (spin-coating)	8

NDI-C₆+C₈-BTBT	0.3	365	-10	4 ms	6 ms	50	1.78	N/A	organic crystalline	9
2-An-BTBT	0.0177	380	-40	20 s	20 s	1576	7136	N/A	single crystal	10
TPA-An	0.0002	370	-60	N/A	N/A	1.03×10^3	7.19×10^5	1.40×10^{16}	single crystal	11
TBA-An			-60	N/A	N/A	3.45×10^4	1.50×10^5	1.60×10^{17}		
C8-BTBT+ polythioether	4.0	350	-30	21 s	16 s	1.0×10^5	2.5	6.3×10^{14}	films (spin-coating)	12
NDI-PM	0.001	365	100	229 ms	268 ms	2.0×10^5	1980	1.8×10^{14}	thin-films	13
BTBT6-syn (Bare SiO₂)	3.1×10^{-5}	370	-60	N/A	N/A	1.2×10^4	9×10^6	2×10^{17}	thin-films	This work
BTBT6-syn (SiO₂/PS)	3.1×10^{-5}	370	-60	N/A	N/A	1.64×10^5	4.5×10^6	3.9×10^{16}		
BTBT6-syn (SiO₂/PMA)	3.1×10^{-5}	370	-60	N/A	N/A	3.47×10^5	1.2×10^6	7.1×10^{16}	thin-films	This work
BTBT6-syn (SiO₂/PPO)	3.1×10^{-5}	370	-60	N/A	N/A	8×10^5	3.6×10^6	2.1×10^{17}		

1. Luo, L. et al. Charge-transfer pentacene/benzothiadiazole derivative cocrystal for UV-to-NIR large range responsive phototransistors. *Org. Electron.* **100**, 106363 (2022).

2. Jiang, L., Huang, H., Gui, F., Xu, Y. & Lin, Q. Ultrasensitive UV-NIR broadband phototransistors based on AgBiS₂-organic hybrid films. *J. Mater. Chem. C*. **9**, 7583-7590 (2021).
3. Zhao, G. et al. High-performance UV-sensitive organic phototransistors based on benzo[1,2-b:4,5-b']dithiophene dimers linked with unsaturated bonds. *Adv. Electron. Mater.* **1**, 1500071 (2015).
4. Li, A. et al. Highly responsive phototransistors based on 2,6-bis(4-methoxyphenyl)anthracene single crystal. *J. Mater. Chem. C*. **5**, 5304-5309 (2017).
5. Huang, J. et al. Printable and flexible phototransistors based on blend of organic semiconductor and biopolymer. *Adv. Funct. Mater.* **27**, 1604163 (2017).
6. Hong, S.H. et al. Photoelectric effect of hybrid ultraviolet-sensitized phototransistors from an n-type organic semiconductor and an all-inorganic perovskite quantum dot photosensitizer. *Nanoscale* **13**, 20498-20507 (2021).
7. Tao, J. et al. Organic UV-sensitive phototransistors based on distriphenylamineethynylpyrene derivatives with ultra-high detectivity approaching 10¹⁸. *Adv. Mater.* **32**, 1907791 (2020).
8. Huang, W., Lin, Y.H. & Anthopoulos, T.D. High speed ultraviolet phototransistors based on an ambipolar fullerene derivative. *ACS Appl. Mater. Interfaces* **10**, 10202-10210 (2018).
9. Guo, J. et al. Few-layer organic crystalline van der waals heterojunctions for ultrafast UV phototransistors. *Adv. Electron. Mater.* **6**, 2000062 (2020).
10. Yan, L. et al. Investigating the single crystal ofet and photo-responsive characteristics based on an anthracene linked benzo[b]benzo[4,5]thieno[2,3-d]thiophene semiconductor. *Org. Electron.* **72**, 1-5 (2019).
11. Tao, J. et al. Organic single crystals with high photoluminescence quantum yields close to 100% and high mobility for optoelectronic devices. *Adv. Mater.* **33**, 2105466 (2021).
12. Peng, H. et al. Interface engineering via photopolymerization-induced phase separation for flexible UV-responsive phototransistors. *ACS Appl. Mater. Interfaces* **10**, 7487-7496 (2018).
13. Song, I. et al. High-performance visible-blind UV phototransistors based on n-type naphthalene diimide nanomaterials. *ACS Appl. Mater. Interfaces* **10**, 11826-11836 (2018).

Comment 2: On page 2, the authors write (see line 62/63):

“Despite the emerging importance of tetrachromatic vision in advanced artificial visual system, relatively little attention has been paid to sensing and processing of UV light.” In case there is a “little attention”, this works should be cited.

Our reply: Thank you for your valuable suggestions. As you can see in the revised manuscript, the corresponding references have been added as Ref.7, Ref.16 and Ref.17.

Our revision: Despite the emerging importance of tetrachromatic vision in advanced artificial visual system, relatively little attention has been paid to sensing and processing of UV light^{7,16,17}.

7. Li, G. et al. Photo-induced non-volatile VO₂ phase transition for neuromorphic ultraviolet sensors. *Nat. Commun.* **13**, 1729 (2022).
16. Shi, J.L. et al. A fully solution-printed photosynaptic transistor array with ultralow energy consumption for artificial-vision neural networks. *Adv. Mater.* **34**, 2200380 (2022).
17. Hong, X.T. et al. Two-dimensional perovskite-gated AlGa_N/Ga_N high-electron-mobility-transistor for neuromorphic vision sensor. *Adv. Sci.* **9**, 2202019 (2022).

REVIEWERS' COMMENTS

Reviewer #1 (Remarks to the Author):

The authors have addressed my comments adequately in the manuscript.

Reviewer #2 (Remarks to the Author):

Thank you to the authors for taking the time to respond to each of my questions and suggestions. I acknowledge that each of the author responses were fair, and that each response has resulted in the appropriate manuscript change. I am happy to see an improvement in the quality of writing and happily surprised to see the authors have recoloured many of their figures.

Reviewer #3 (Remarks to the Author):

Publish as it is.

Response to Reviewer #1

Comment: The authors have addressed my comments adequately in the manuscript.

Our reply: We highly appreciate your professional and constructive comments, which guided us to improve the manuscript.

Response to Reviewer #2

Comment: Thank you to the authors for taking the time to respond to each of my questions and suggestions. I acknowledge that each of the author responses were fair, and that each response has resulted in the appropriate manuscript change. I am happy to see an improvement in the quality of writing and happily surprised to see the authors have recoloured many of their figures.

Our reply: We sincerely appreciate your valuable comments and suggestions that greatly helped us to improve the quality of this manuscript.

Response to Reviewer #3

Comment: Publish as it is.

Our reply: Thank you for the positive comments for our work.